# Seasonal Dynamics and Damage of *Halyomorpha halys* in Italian Vineyards

**DOI:** 10.3390/insects15060378

**Published:** 2024-05-22

**Authors:** Davide Scaccini, Diego Fornasiero, Paola Tirello, Simone Vincenzi, Massimo Cecchetto, Ilirjan Allgjata, Carlo Duso, Alberto Pozzebon

**Affiliations:** Department of Agronomy, Food, Natural Resources, Animals and Environment, University of Padova, Viale dell’Università 16, Legnaro, 35020 Padova, Italy; diego281m@hotmail.it (D.F.); paola.tirello@unipd.it (P.T.); simone.vincenzi@unipd.it (S.V.); massimo96cec@gmail.com (M.C.); ilirjan93all@gmail.com (I.A.); carlo.duso@unipd.it (C.D.)

**Keywords:** brown marmorated stink bug, *Vitis vinifera*, grapevine, cultivars, *Botrytis cinerea*, grey mold, sour rot, pathogen, IPM, Italy

## Abstract

**Simple Summary:**

The brown marmorated stink bug, *Halyomorpha halys* (Stål) (Hemiptera: Pentatomidae), is an invasive pest commonly found in vineyards. This study aimed to investigate the seasonal patterns and distribution of *H. halys* in vineyards with multiple grape cultivars. We also assessed the damage caused by different levels of pest infestation on grape clusters. Our findings revealed that *H. halys* populations varied throughout the seasons and were higher on red berry cultivars, such as Cabernet Franc and Merlot, and those ripening later in the season. Additionally, we observed higher pest infestations at the vineyard borders compared to their centers. The study showed that *H. halys* infestations led to damage to grape berries, with varying susceptibility among different cultivars and phenological stages. Furthermore, the presence of *H. halys* increased the incidence on clusters of *Botrytis cinerea* and sour rot, highlighting its significant impact on grapevine health.

**Abstract:**

The brown marmorated stink bug, *Halyomorpha halys* (Stål) (Hemiptera: Pentatomidae), is an invasive polyphagous pest often observed in vineyards. In Europe, a gap needs to be filled in the knowledge on *H. halys* seasonal dynamics and damage on grapes. With this study, we described the seasonal dynamics of *H. halys* and its distribution in multi-cultivar vineyards, and we evaluated the damage on grape clusters induced by different pest densities. In vineyards, the seasonal occurrence of *H. halys* varied across time and grape cultivars, and the pest was more abundant on Cabernet Franc, Merlot and, to a lesser extent, Pinot gris. Moreover, higher densities of *H. halys* were found on red berry cultivars than on white ones, and on cultivars ripening late in the season. An edge effect was also detected in pest distribution within vineyards, with more stink bugs observed in the borders. In the study on pest infestation density, *H. halys* caused damage on berries, showing differences in susceptibility among different cultivars and with regard to the time of infestation (i.e., plant phenological stages). *Halyomorpha halys* infestation induced an increase in *Botrytis cinerea* and sour rot incidence, which probably represents the main issue related to the impact of brown marmorated stink bug on grapevine.

## 1. Introduction

In terms of cultivated surface and economic significance, Europe stands among the world’s top producers of grapes, with the wine grape, *Vitis vinifera* L. (Vitales: Vitaceae) being one of the major fruit crops [1]. Worldwide, Italy is one of the most important countries for wine grape production [2,3], where vineyards cover 718,000 ha, nearly 10% of the global vineyard area [1]. The Veneto region (northern Italy) is one of the main vineyard areas of Italy, producing over 1,500,000 Mg and ranking among the top regions in Italy for grapevine production [4].

The brown marmorated stink bug, *Halyomorpha halys* (Stål) (Hemiptera: Pentatomidae), is a polyphagous pest that can feed on more than 170 host plant species including grapevine [5], on which it has also been recorded to feed in Europe [6,7,8]. The presence of all developmental stages of *H. halys* in vineyards suggests that *V. vinifera* may serve as reproductive host for this pest [9]. However, laboratory and semi-field studies indicate that nymphs are able to feed on grapevine but unable to complete their development on this host plant [10,11]. In the US, *H. halys* is regarded as a potential minor pest for grape production due to its low preference toward *V. vinifera* compared to other plant species [12].

Particularly in early spring, following their gradual emergence from overwintering shelters, adults start searching for host plants and colonizing cultivated ones. In Europe, this pest can undergo 1–2 generations per year, mostly in summer, and there is a coexistence of various developmental stages that typically peak in numbers in August–September [13]. During its period of activity, *H. halys* is capable of long-distance movements, with adults that can disperse several kilometers by flying [14,15], and nymphs that can walk between plots at farm scale [16]. *Halyomorpha halys* is a pest known to take advantage of mixed-host diets, and these feeding habits can increase nymphal survivorship and hasten developmental duration [11,17]. This pest also exhibits a spatial distribution pattern that is influenced by the habitat composition of agricultural landscapes. This includes movements related to the presence of plants and the ripening of fruits during various periods of the growing season, encompassing different crops and non-crop plant species [18,19,20,21,22].

The damage caused by an *H. halys* infestation usually derives from the feeding activity of both immature and adult stages [5], sometimes leading to an increased incidence of plant diseases [23,24,25,26,27]. *Halyomorpha halys* exhibits a preference for plant fruits such as legumes, drupes, nuts, and berries [5], but can feed on stems and trunks [28,29]. On grapevines, *H. halys* feeding activity on grape clusters results in discoloration, necrosis, softening of berries, and their drop [12,30], possibly increasing pathogens and other pest infestations as it does in other crop species [5,31]. Furthermore, in the case of high brown marmorated stink bug infestations on grapevine clusters in field or post-harvest structures, there is a risk of *H. halys* taint, which can impact the aroma and taste of musts [32,33,34,35,36]. However, this issue appears to be less pronounced when the infestation levels of the pest are low [36]. In spite of being uncommon [7,37], cases of high infestations in the field have been recorded in vineyards in the US, Russia and Georgia, especially in mid- and late-summer [9,19,31,38], but the effect on grapevine cultivation was frequently overlooked.

Studies specifically focused on pest population dynamics and damage potential on grapevines are often scattered across other regions of invasion, even in those particularly suitable for viticulture. The aims of this study were as follows: (1) to describe the seasonal dynamics of *H. halys* in vineyards comprising different grape cultivars and their spatiotemporal distribution in the vineyard; and (2) to evaluate the damage on grape clusters and musts (grape juice) by different infestation densities of *H. halys*.

## 2. Materials and Methods

### 2.1. Seasonal Dynamics of *H. halys* in the Vineyards

The seasonal dynamics and distribution of *H. halys* were studied in four multi-cultivar vineyards located in the Veneto region (north-eastern Italy) that were surveyed every 14 days from July to September (2017) or 7 days from June to September (2018). Sampling timing was more frequent during the second year based on the findings from the first year. Vineyards were managed according to common practices in the Veneto region with application of fungicides and insecticides against the main grape pathogens and pests, and with grass cover constantly maintained.

The sampled grapevine cultivars represented the main ones in the region [39], namely Cabernet Franc, Cabernet Sauvignon, Merlot and Raboso (red berry), and Glera, Pinot gris and Tai (white berry). The cultivars Cabernet Franc, Cabernet Sauvignon and Raboso were sampled only in 2018. Surveys were conducted using both visual (2017) and beat samplings (2017 and 2018). Sampling points were at both the border and the center of the vineyard rows, each comprising four plants. In each plot, sampling was carried out at a minimum of four locations, resulting in the sampling of at least 16 plants per grape variety and date. During visual sampling, the canopy of plants was inspected for a period of three minutes on four plants at each location, and *H. halys* adults and nymphs were counted. Beat sampling was also performed on the same plants by shaking the canopy and collecting insects on a tray (white sheet, 1 m × 1 m, UM01–Clap Net, Omnes Artes s.a.s., Bergamo, Italy) placed below, with counts made for *H. halys* nymphs and adults.

### 2.2. Assessment of *H. halys* Damage to the Grape Cluster

In two vineyards located in the Veneto region (north-eastern Italy), we established controlled infestations of *H. halys* adults by enclosing grape clusters within insect-proof net cages on four cultivars covering both red and white grapes (two cultivars each): Cabernet Franc and Merlot (first vineyard), and Glera and Pinot gris (second vineyard). The cages consisted of ~55 L (Ø 30 cm × 80 cm h) insect-proof, white netting with a mesh of 150 µm, installed on the plants early in the season. Each cage included two clusters, and four infestation densities were established: 0 (control), 0.5, 1, or 2 *H. halys* adults per cluster. The infestation lasted 10 days, and each treatment was replicated six times. We replicated the infestation treatment on different clusters during the season. Specifically, infestations were performed following different grapevine phenological stages: bunch closure (BBCH 77), starting veraison (BBCH 81), 50% of veraison (BBCH 83), starting ripening (BBCH 85), and 100% ripened (BBCH 89). Deceased adults were substituted with live ones to maintain a constant infestation level. After the infestation period, adults were removed and the cages were kept on the plant until harvesting, when the final assessment was performed in the laboratory, i.e., the presence of berries with signs of *H. halys* infestation and the mean weight of berries for each cluster. Berries with infestation signs were divided into two categories: (i) those that showed feeding damage by *H. halys* (fruit presenting discoloration, necrosis, and softening of the berry [31]), and (ii) those with signs of grey mold, *Botrytis cinerea* (Pers.) (Helotiales: Sclerotiniaceae) and sour rot infection.

Furthermore, grape clusters were manually pressed and the obtained must was analyzed after adding 1 g L^−1^ of potassium metabisulfite for preservation. Then, further analyses were performed on glucose, fructose, tartaric and gluconic acid content through High-Performance Liquid Chromatography (HPLC). Gluconic acid was analyzed to assess the occurrence of *B. cinerea*, as it is one of its secondary metabolites that indicates grey mold infestations [40,41,42,43,44]. This method enhances the precision of grey mold evaluation compared to visual estimations, which may introduce bias [45].

### 2.3. Fungicide and Insecticide Application on *H. halys* Infestation and Grey Mold and Sour Rot Infection

During 2019, an experiment was performed on the effect of insecticide and fungicide applications in reducing *H. halys* infestation and pathogen infection in a vineyard (cv. Pinot gris) located in Belfiore (Verona, Italy). The experiment followed a complete two-factorial design where the experimental factors were as follows: (a) insecticide application [three levels: Control, Acetamiprid (active ingredient 4.67%, dose 1.5 L ha^−1^, Epik^®^ SL Sipcam Italia S.p.A., Milan, Italy) and Deltamethrin (active ingredient 2.42%, dose 0.7 L ha^−1^, Decis Evo^®^ Bayer CropScience S.r.l., Milan, Italy)]; and (b) fungicide application [two levels: Control and Fungicide (active ingredients: Eugenol 3.2%, Geraniol 6.4%, Timolol 6.4%; dose 4 L ha^−1^, 3LOGY^®^ Sipcam Italia S.p.A., Milan, Italy)]. The experiment comprised six treatments with four replicates of 40 plants each. Insecticide and fungicide applications were performed on July 29th. Stink bug infestation was evaluated before the pesticide spraying and after 3 and 7 days, using the beat sampling technique to collect insects on a tray as described for the first experiment. Samplings were performed on four plants located in the central part of each replicate. Moreover, at harvest, the number of berries exhibiting signs of grey mold and sour rot infection was assessed on 25 clusters randomly collected in the central part of each replicate.

### 2.4. Statistical Analyses

Data on brown marmorated stink bug seasonal dynamics were analyzed with generalized linear mixed repeated measures models. *Halyomorpha halys* individuals collected through beat sampling (in 2017 and 2018) or observed during visual sampling (in 2017) were treated as a dependent variable with repeated measures, corresponding to different sampling dates. During visual sampling monitoring, a high number of zero-data occurrences were observed. Consequently, the data were analyzed using the GLIMMIX procedure of SAS, version 9.4 [46], with a Poisson distribution. In this analysis, cultivars, sampling time, position within vineyard rows (i.e., border vs. center), and their interactions were considered as fixed effects and tested with an F-test (α = 0.05). While modeling, different vineyards were considered as random effect terms. In this analysis, untransformed data were used, and model assumptions were evaluated by inspecting diagnostic plots of model residuals.

Data on *H. halys* monitoring through beat sampling were analyzed with a generalized linear mixed repeated measures model with the MIXED procedure of SAS, ver. 9.4 [46]. In this analysis, cultivars, sampling time, position within vineyard rows, and their interactions were considered as fixed effects and tested with an F-test (α = 0.05). Data from the two seasons were analyzed separately. While modeling, different vineyards were considered as random effect terms. Tukey’s test on the least-square means (α = 0.05) was used post hoc to evaluate differences among cultivars and between the positions within vineyard rows. In data analysis of the second growing season sampling, when using an F-test (α = 0.05), contrasts were also designed to compare *H. halys* numbers between white and red berry cultivars (i.e., Glera, Pinot gris and Tai vs. Cabernet Franc, Cabernet Sauvignon, Merlot and Raboso), and cultivars categorized according to their harvesting periods (i.e., early: Pinot gris and Tai; medium: Cabernet Sauvignon, Glera and Merlot; or late season: Cabernet Franc and Raboso). Data were checked for model assumptions prior to the analysis and were log (x + 1) transformed before the statistical analysis.

Data obtained from the cage experiment (i.e., weight of the cluster, number of berries per cluster, berry weight, and percentage of berries per cluster with signs of *H. halys* feeding damage or with signs of *B. cinerea*) were analyzed through a general linear mixed repeated measures model with the MIXED procedure of SAS, ver. 9.4 [46]. Cultivars, infestation density, phenological stage, and their interactions were considered as fixed effects and tested using F-tests (α = 0.05) followed by a post hoc Tukey–Kramer’s test (α = 0.05). The farm was included as a random factor. Data obtained from the laboratory HPLC were analyzed through the same model as reported before. In each pair, red and white cultivars were analyzed separately (i.e., merging Cabernet Franc and Merlot, and Glera and Pinot gris). Transformed data were used prior to the statistical analysis as arcsin √x in the case of percentage, and as log (x + 1) for the others.

Data from the experiment on fungicide and insecticide application were analyzed with a general linear mixed repeated measures model with the MIXED procedure of SAS, ver. 9.4 [46]. Treatment, time, and their interaction were considered as fixed effects in the analysis and tested with an F-test (α = 0.05). *Halyomorpha halys* individuals by beat sampling were considered as a dependent variable made with repeated measures (i.e., before spraying, and 3 and 7 days after spraying). While modeling, different vineyards were considered as random effect terms. Tukey’s test on the least-square means (α = 0.05) was used post hoc to evaluate differences among treatments. Data were checked for model assumptions prior to the analysis and were log (x + 1) transformed before the statistical analysis.

Data on pathogens infestation on Pinot gris berries were analyzed through a general linear mixed repeated measures model with the MIXED procedure of SAS, ver. 9.4 [46], considering the treatment (i.e., not treated, treated with insecticide and/or fungicide) as fixed effects in the model, and tested using F-tests (α = 0.05) followed by a post hoc Tukey–Kramer’s test (α = 0.05). When modeling, different vineyards were considered as random effect terms. Transformed data were used prior to the statistical analysis as arcsin √x on the rate of damaged berries per cluster.

## 3. Results

### 3.1. Seasonal Dynamics of *H. halys* in the Vineyards

During the two growing seasons, all life stages of *H. halys* were observed in vineyards. In 2017, beat sampling revealed a higher count of stink bugs if compared to visual sampling, with the latter consistently underestimating their numbers (Figure 1A and Appendix A). Using beat sampling, a variation in *H. halys* numbers was observed during the season (Figure 1A). The infestation level was also influenced by the grape cultivar (Table 1), with higher levels on Merlot and Pinot gris compared to Cabernet Franc and Glera (Figure 2A).

During the 2018 growing season, the *H. halys* population levels fluctuated during the samplings (Table 1), increasing at the end of the season in Cabernet Franc, and Raboso (Figure 1B). Overall, we observed differences among cultivars, with Cabernet Franc and Merlot showing the highest infestation levels, while Tai, Glera, Raboso and Cabernet Sauvignon the lowest (Figure 2B). Pinot gris experienced a moderate level of infestation. *Halyomorpha halys* numbers were higher on red grape cultivars than on white ones (Figure 3A), and in late-ripening cultivars compared to early ones (Table 2; Figure 3B). Pest distribution along rows was influenced by the position, being higher at the border than at the center (Table 1; Figure 3C).

### 3.2. Assessment of *H. halys* Damage to the Grape Cluster

In the cage experiment, feeding damage on clusters increased with the increase in *H. halys* infestation level in all cultivars (Table 3). Concerning white cultivars, a significant interaction among all experimental factors was found on the percentage of berries showing signs of infestation (Table 3). On Glera, the presence of damaged berries was different among infestation density and phenology, exhibiting the highest damage in the treatment with two *H. halys* adults per cluster at 50% veraison (Figure 4). In contrast, on Pinot gris, a similar level of damage was observed only in treatments with *H. halys* infestation, independently of phenological stages (Figure 4). On red cultivars, the percentage of damaged berries per cluster showed a general increase along with the increase in infestation density, and a higher rate of damaged berries were observed on Cabernet Franc than Merlot (Figure 5). On red cultivars, the effect of infestation was independent of the phenological stages (Table 3).

The percentage of berries with signs of *B. cinerea* and sour rot due to brown marmorated stink bug infestations was different between white-berry cultivars (Table 4) but not for red-berry ones, and the incidence of this pathogen was higher on Pinot gris than on Glera (Figure 6A). In red cultivars, signs of the pathogen were more represented on Merlot than on Cabernet Franc (Table 4; Appendix A). On white cultivars, the presence of signs of grey mold and sour rot associated with *H. halys* infestation differed among phenological stages, with a higher incidence following infestations from the grape ripening period (Figure 6B).

An effect of brown marmorated stink bug infestation on the mean berry weight emerged on red cultivars but not on white ones (Table 5). On red cultivars, indeed, berry weight was higher in the case of 0.5 and 1 *H. halys* adults per cluster than in the other two infestation densities (Figure 7A), with differences in the interaction cultivar × plant phenological stage (Figure 7B).

In the must, no effect of *H. halys* infestation was observed on glucose concentration, with differences solely related to the cultivar features (Table 6; Appendix A). An increase in fructose concentration was observed in infested berries of white cultivars, but not in red ones (Table 6; Figure 8).

A reduction in the tartaric acid concentration in the must obtained from white cultivars was observed in Pinot gris and limited to treatments with *H. halys* infestation at the start of veraison. In contrast, no differences emerged on Glera (Table 7; Figure 9). In red cultivars, infestation by brown marmorated stink bug was associated with an increase in tartaric acid concentration (Table 7; Figure 10A). Other differences were associated with cultivars and phenological stages (Table 7; Figure 10B).

In both white and red cultivar musts, gluconic acid concentration was influenced by *H. halys* infestation level and the phenological stage when this infestation occurred, with a significant interaction observed in the case of white grape cultivars (Table 7; Figure 11). In both white and red grape cultivars, gluconic acid increased along with *H. halys* infestation, and this increase was associated with infestation established at the beginning of ripening in white-berry cultivars (Table 7; Figure 11 and Figure 12).

### 3.3. Fungicide and Insecticide Application on *H. halys* Infestation and Grey Mold and Sour Rot Infection

Results from the experiment with applications of fungicide and insecticides showed differences in *H. halys* infestation at three and seven days after treatments (Table 8). Higher stink bug numbers were observed on control plots and those treated with fungicide only as compared to those receiving insecticide applications (Table 8; Figure 13).

The percentage of grape berries showing signs of grey mold and sour rot infection was influenced by treatment (F5, 42 = 2.84; *p* = 0.0268) and was higher in the control than all the other treatments (Figure 14).

## 4. Discussion

This study shows that *H. halys* can infest vineyards in northern Italy, and its seasonal dynamics and infestation level can change among grape cultivars. Previous studies from North America found that populations of brown marmorated stink bug in vineyards often fluctuate in time and space [9,47]. We found that in north Italian vineyards, the brown marmorated stink bug showed a preference for some cultivars like Cabernet Franc, Merlot and, to a lesser extent, Pinot gris. However, while Cabernet Franc and Merlot exhibited the highest infestation levels in 2018, the first showed a lower infestation in 2017 compared to Merlot and Pinot gris. Variations in infestation levels between years may be due to changes in volatiles emitted by the crop [48,49], possibly induced by abiotic factors such as water conditions and temperature [50,51]. Nonetheless, some varieties consistently remained more susceptible to *H. halys* infestation in both years.

While *V. vinifera* is acknowledged as a potential host plant for *H. halys* [9], it is not regarded as the preferred host plant [52]. Nevertheless, we consistently observed different developmental stages of the pest occurring throughout the sampling period in the conditions under investigation, indicating a certain level of plant suitability. The preference for host plants is likely influenced by the availability of other plant species in the vicinity. This phenomenon was observed in the US, where various plant species, including grapevines, were identified as preferred hosts, resulting in high numbers of *H. halys* [19]. On the other hand, in other studies, brown marmorated stink bug infestations on grapevines were found to be less severe, indicating that it may be a minor grapevine pest under the conditions they investigated [12].

In our study, pest infestation was higher at field edges than in the center, which is known as border field infestation and is already known in row crops [53,54,55,56,57], but also in vineyards and orchards [9,47,58,59]. The inability of the pest to complete its development solely by feeding on grapes [10,11] could have contributed to the increased infestation level at the border observed in the surveyed vineyards. This information can be useful for *H. halys* management, such as IPM-CPR (Integrated Pest Management—Crop Perimeter Restructuring) practices that are known to reduce the use of insecticides while reducing the pest damage [60].

In vineyards, higher *H. halys* infestation levels were observed on red grape cultivars compared to white, as well as on cultivars with a late ripening compared to early ripening ones. Brown marmorated stink bug numbers, particularly adults, typically increase when fruit ripens on crops [5,23,61]. This was consistent throughout the seasons in 2017, while in 2018 an increase was observed in the second part of the season, particularly in red cultivars (Cabernet Franc, Merlot, and Raboso). It is known that *H. halys* can feed on non-reproductive parts of the plant [28,29], but prefers to feed on ripening fruits [62,63,64,65]. Indeed, during sampling, we observed insects on ripening berries, while in earlier phases of the season, they were more distributed on the plant canopy (D. Scaccini and A. Pozzebon, pers. obs.). In northeastern Italy, the grapevine is the primary perennial crop, and a significant portion of the landscape is dominated by vineyards. Additionally, grapevines ripen in late summer and fall, while other crops susceptible to *H. halys* ripen from spring to late summer, potentially impacting stink bug movement and host plant selection. In landscapes dominated by vineyards, grape berries could serve as a primary food source for brown marmorated stink bug during the ripening season, and this study reported an increase in pest numbers at the end of the growing season.

High infestations of *H. halys* can damage berries, causing them to become soft and show necrotic spots and discolored parts on the skin [31,37,47], and different grape cultivars seemed to be more susceptible to *H. halys* than others, as in the case of Glera compared to Pinot gris, or Cabernet Franc compared to Merlot. Differences among cultivars in terms of susceptibility to *H. halys* were also observed in the US, where the number of feeding punctures on Seyval Blanc was higher than those recorded on Cabernet Sauvignon berries [47]. The amount of damage seemed to be related to the phenological stage of the plant. Previous records of damage on *V. vinifera* berries indicated that feeding punctures were more common during pre-harvest and veraison stages, with more pronounced injuries during veraison [47]. Here, we found a higher susceptibility in terms of direct feeding damage when infestations occurred during 50% of the veraison, which represents the phenological stage when *H. halys* infestation level may increase in the field. The damage caused by brown marmorated stink bug led to certain consequences in terms of berry weight and sugar and acid concentration in the must, but these parameters were not significantly compromised. The levels of glucose and fructose appeared to be more strongly associated with the grape cultivar or its phenological stage rather than the infestation of *H. halys*. It is known that fructose levels remain unchanged in the fruit after *B. cinerea* inoculation, while glucose levels can either increase or decrease depending on the abiotic environment [66], thus making the influence of pathogen occurrence unclear. Besides direct feeding damage, *H. halys* infestation was associated with an increase in the incidence of *B. cinerea* and sour rot, particularly on Pinot gris. These differences were supported by the analysis of gluconic acid, which is associated with grey mold infections and was higher in treatments with clusters infested by *H. halys* individuals for both white and red cultivars. Furthermore, the incidence of grey mold and sour rot increased in Pinot gris plants that were left untreated by insecticides or fungicides. Lower signs of grey mold and sour rot infection were observed in fungicide and insecticide treated plants, with no differences between the application of insecticide and fungicide solely or in combinations. These findings confirm the association between stink bug infestation and pathogen incidence. Moreover, they underscore that the application of fungicides—which target pathogens directly—or insecticides—which target the pest—leads to a reduction in pathogen damage incidence. It should be stressed that the natural incidence of the pathogens was not particularly high in the control (20%), and the applications of the fungicide and insecticides were associated with a reduction of 50% in the damaged berries rate (about 10%).

From the cage study, it emerged that the period of infestation by *H. halys* can also play a role in the induction of grey mold infection, with an increase in gluconic acid concentration after infestation during the start of ripening. This relation may lead to facilitation of pathogen infestation in the case of pest presence, likely due to the opening of wounds in the skin of berries. It is undeniable that Pentatomidae may transmit pathogens, especially fungi [67,68]; however, specific studies on the plant–pathogen system involving the brown marmorated stink bug are still lacking. Of the few available, on cherries, *H. halys* infestations increased the number of fruits with fungal infections [26], and on fruits and vegetables, this pest was able to transmit yeasts [69], as well as other fungi and bacteria [23,24,25,70]. The increase in grey mold infection should thus be considered as a threat to grapevine production. It should be mentioned that the vineyards where the experiments were performed were subject to standard application of fungicides, and specific management against grey mold was applied. Moreover, the occurrence of *B. cinerea* and sour rot infections can be influenced by climatic conditions during the ripening period. At the time of the experiments, conditions were not particularly conducive to grey mold development, so we can expect a higher incidence when more favorable conditions occur. In addition, the association between the incidence of *B. cinerea* and sour rot and *H. halys* infestation seems not to be linear, with an increase in pathogen incidence also when low pest infestation occurs.

## 5. Conclusions

In conclusion, our results showed that *H. halys* can infest vineyards and that its density is higher at the margins of the field than in the center. Pest infestation density was influenced by grapevine cultivars and their phenological stage. However, from the cage experiment, we found that direct feeding damage is relevant only at high infestation levels. Notably, infestation levels like those that cause damage to grape clusters were not observed in field conditions nor at vineyard margins. However, not only direct damage may affect grapevine production since we found secondary damage related to the increase in pathogen incidence following stink bug infestation. This has important implications in the management of grapevine diseases and can be the main issue related to *H. halys* infestation on grapevine.

## Figures and Tables

**Figure 1 insects-15-00378-f001:**
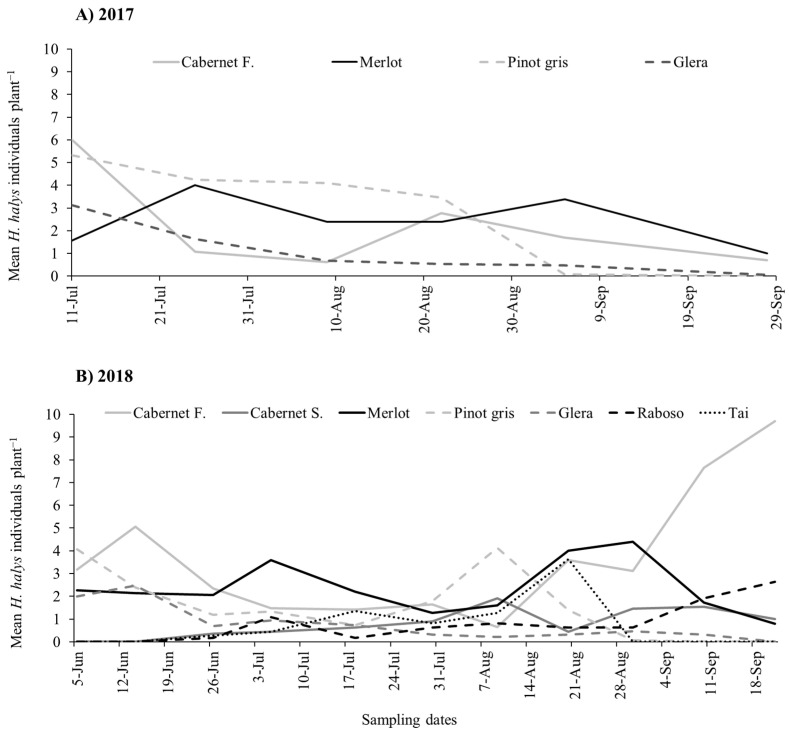
*Halyomorpha halys* seasonal dynamics on different grape cultivars observed from beat sampling in 2017 (**A**) and 2018 (**B**).

**Figure 2 insects-15-00378-f002:**
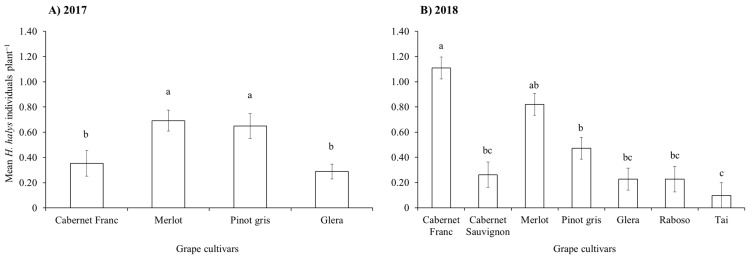
*Halyomorpha halys* number (mean ± std. err.) on different grape cultivars observed from beat sampling in 2017 (**A**) and 2018 (**B**). Back-transformed data from least-square means were used in the figure. Different letters indicate significant differences in the Tukey’s test on least-square means (α = 0.05).

**Figure 3 insects-15-00378-f003:**
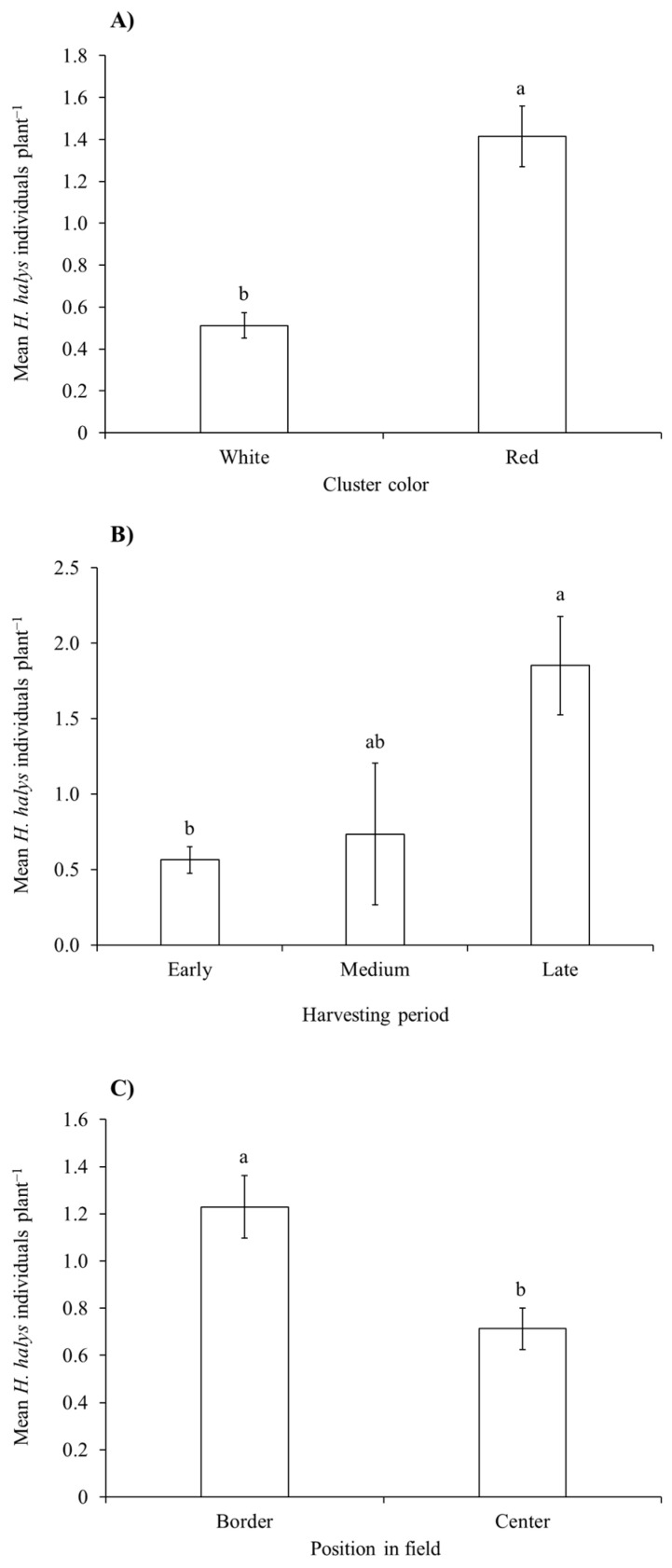
*Halyomorpha halys* abundance (mean no. ± std. err.) in vineyards on white and red grape cultivars (**A**), on cultivars with different harvesting period (**B**), and in different positions (i.e., border vs. center) within vineyard rows (**C**) as observed in 2018. Back-transformed data from least-square means were used in the figure. Different letters indicate significant differences in the Tukey’s test on least-square means (α = 0.05).

**Figure 4 insects-15-00378-f004:**
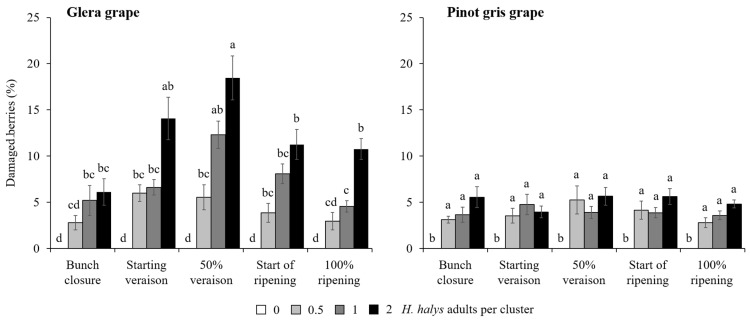
Percentage (mean ± std. err.) of berries of white cultivars showing signs of feeding damage observed in the cage experiment in different treatments defined by *Halyomorpha halys* infestation density and phenological stage of infestation. Different letters indicate significant differences according to Tukey–Kramer’s test (α = 0.05).

**Figure 5 insects-15-00378-f005:**
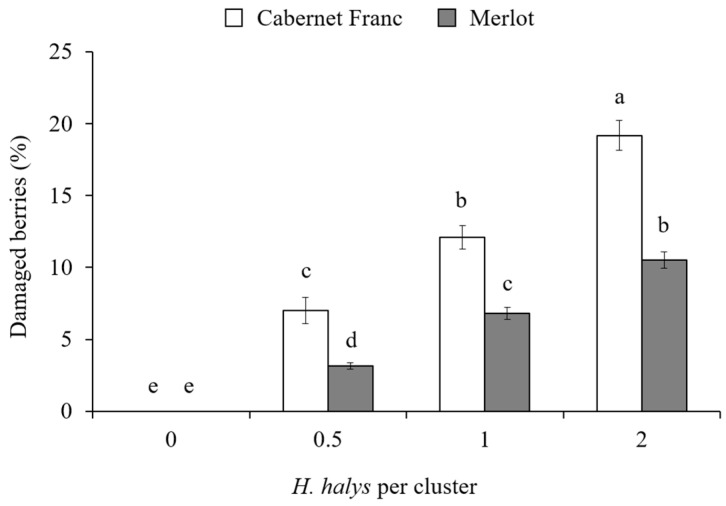
Percentage (mean ± std. err.) of damaged berries by *Halyomorpha halys* per cluster of the red grapes Cabernet Franc and Merlot with different *H. halys* infestation densities. Different letters indicate significant differences according to Tukey–Kramer’s test (α = 0.05).

**Figure 6 insects-15-00378-f006:**
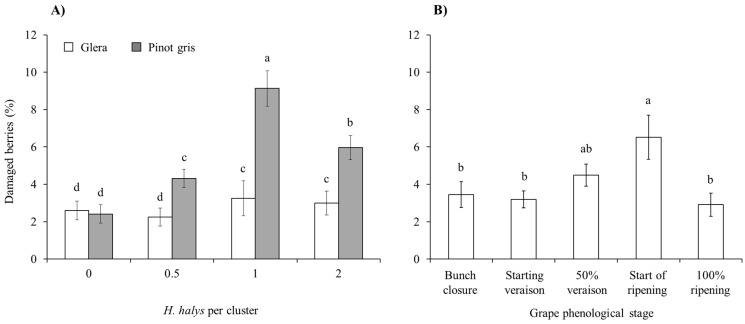
Percentage (mean ± std. err.) of berries with signs of *Botrytis cinerea* and sour rot in the white grapes Glera and Pinot gris with different *Halyomorpha halys* infestation densities (**A**), and for different plant phenological stages (**B**). Different letters indicate significant differences according to Tukey–Kramer’s test (α = 0.05).

**Figure 7 insects-15-00378-f007:**
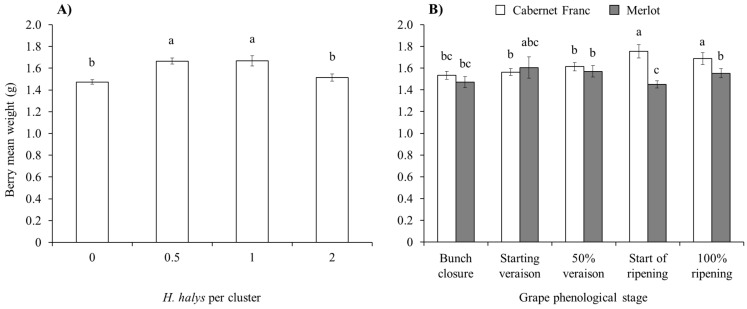
Berry weight (mean ± std. err.) in red grapes for different *Halyomorpha halys* densities (**A**), and considering the phenological stage of Cabernet Franc and Merlot (**B**). Different letters indicate significant differences according to Tukey–Kramer’s test (α = 0.05).

**Figure 8 insects-15-00378-f008:**
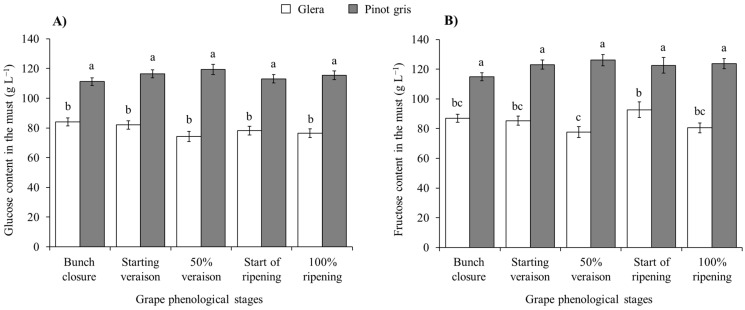
Glucose (**A**) and fructose (**B**) content (mean ± std. err.) in must of white grapes for different *Halyomorpha halys* densities. Different letters indicate significant differences according to Tukey–Kramer’s test (α = 0.05).

**Figure 9 insects-15-00378-f009:**
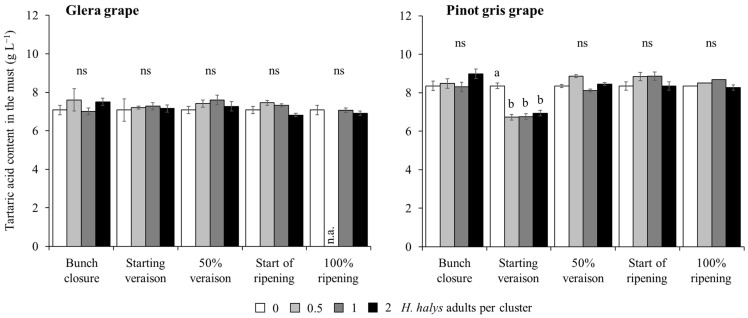
Tartaric acid content (mean ± std. err.) in must of white grapes, Glera and Pinot gris, according to the plant phenological stage and density of *Halyomorpha halys* infestation. Different letters indicate significant differences according to Tukey–Kramer’s test (α = 0.05). n.a. = not available data.

**Figure 10 insects-15-00378-f010:**
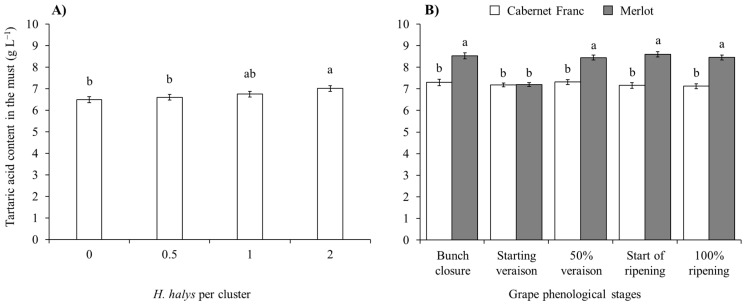
Tartaric acid content (mean ± std. err.) in must of red grapes for different *Halyomorpha halys* densities (**A**), and depending on the phenological stage of Cabernet Franc and Merlot grapes (**B**). Different letters indicate significant differences according to Tukey–Kramer’s test (α = 0.05).

**Figure 11 insects-15-00378-f011:**
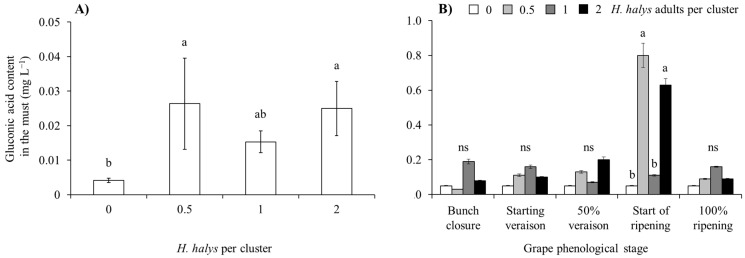
Gluconic acid content (mean ± std. err.) in must of white grapes for different *Halyomorpha halys* densities (**A**), and in different grape phenological stages and *H. halys* infestation densities (**B**). Different letters indicate significant differences according to Tukey–Kramer’s test (α = 0.05).

**Figure 12 insects-15-00378-f012:**
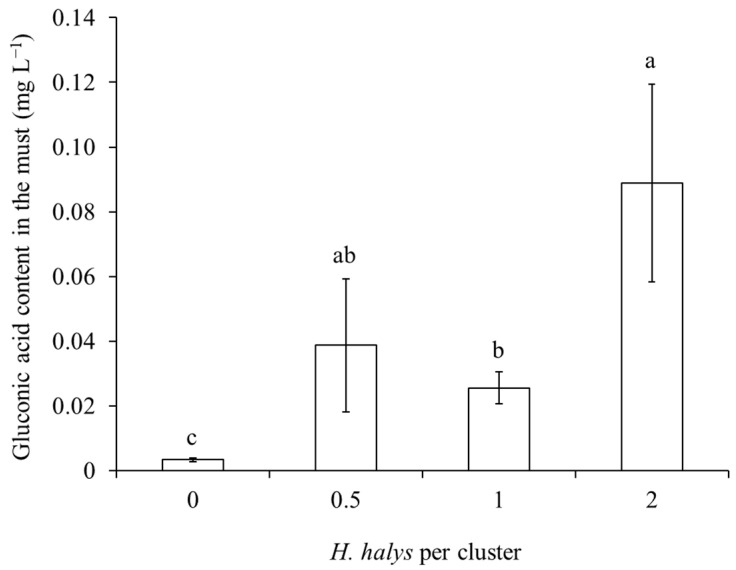
Gluconic acid content (mean ± std. err.) in must of red grapes for different *Halyomorpha halys* densities. Different letters indicate significant differences according to Tukey–Kramer’s test (α = 0.05).

**Figure 13 insects-15-00378-f013:**
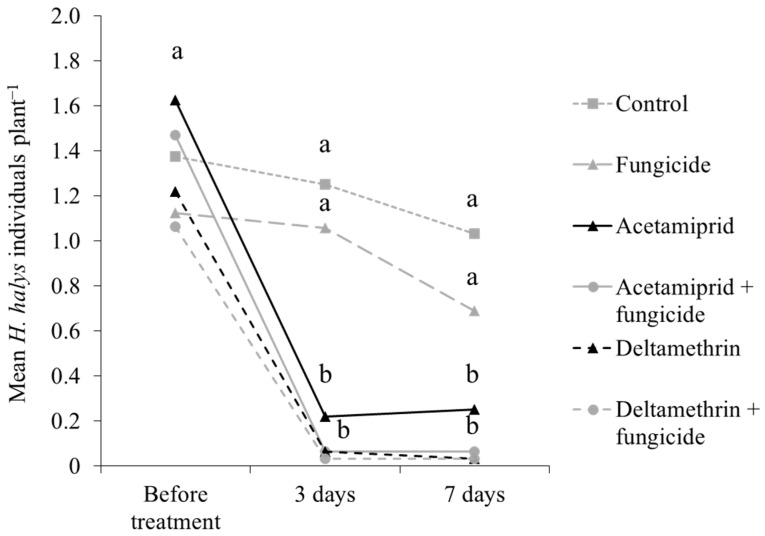
*Halyomorpha halys* number (mean ± std. err.) on Pinot gris grapes collected with beat sampling in 2019, by insecticide/fungicide treatment. Different letters indicate significant differences according to Tukey’s test on least-square means (α = 0.05), by sampling event.

**Figure 14 insects-15-00378-f014:**
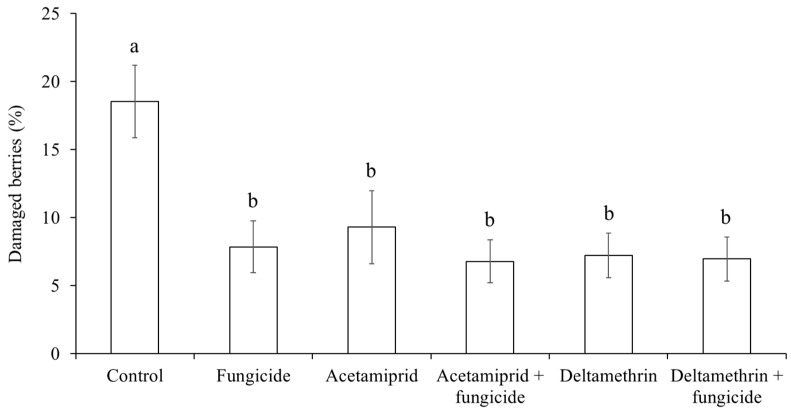
Percentage (mean ± std. err.) of Pinot gris berries with signs of *Botrytis cinerea* and sour rot in grapes observed at harvest in different treatments of fungicide and insecticide applications. Different letters indicate significant differences according to Tukey–Kramer’s test (α = 0.05).

**Table 1 insects-15-00378-t001:** Results of GLMM model (α = 0.05) on the seasonal dynamic of *Halyomorpha halys* observed in vineyards in 2017 and 2018.

Sampling Method	Source of Variation	df	F Value	*p* Value
Visual sampling (2017)	Sampling date (D)	5, 444	0.24	0.9440
	Position (Pos)	1, 427.7	0.99	0.3202
	Cultivar (Cv)	9, 427.8	0.91	0.5187
	D × Pos	5, 444.3	0.92	0.5812
	D × Cv	45, 444.2	1.01	0.4613
	Pos × Cv	5, 428.2	0.65	0.6621
	D × Pos × Cv	25, 444.5	0.92	0.5812
Beat sampling (2017)	D	4, 216	3.94	0.0042
	Pos	1, 106	0.06	0.8075
	Cv	3, 103	3.49	0.0185
	D × Pos	4, 216	0.46	0.7614
	D × Cv	12, 230	1.35	0.1903
	Pos × Cv	3, 103	0.89	0.4515
	D × Pos × Cv	12, 230	0.57	0.8663
Beat sampling (2018)	D	10, 339	2.32	0.0119
	Pos	1, 79.1	7.06	0.0096
	Cv	6, 79.1	7.36	<0.0001
	D × Pos	10, 339	0.22	0.9944
	D × Cv	60, 345	1.81	0.0006
	Pos × Cv	6, 79.2	0.34	0.9152
	D × Pos × Cv	60, 345	0.56	0.9960

**Table 2 insects-15-00378-t002:** Results of contrasts on the seasonal dynamics of *Halyomorpha halys* in vineyards observed in 2018.

Source of Variation	df	F Value	*p* Value
Red vs. White	1, 79.1	10.54	0.0017
Early vs. Medium	1, 79.1	1.66	0.2008
Early vs. Late	1, 79.1	7.22	0.0088
Medium vs. Late	1, 79.1	2.83	0.0965

**Table 3 insects-15-00378-t003:** Results of GLMM (α = 0.05) on the percentage of berries with symptoms of *Halyomorpha halys* feeding on white and red cultivars.

Cultivar Category	Source of Variation	df	F Value	*p* Value
White cultivars	Cultivar (Cv)	1, 440	52.50	<0.0001
	Phenological stage (Phen)	4, 440	8.75	<0.0001
	*H. halys* density (Dens)	3, 440	319	<0.0001
	Cv × Phen	4, 440	6.13	<0.0001
	Cv × Dens	3, 440	18.6	<0.0001
	Phen × Dens	12, 440	1.53	0.1094
	Cv × Phen × Dens	12, 440	2.08	0.0174
Red cultivars	Cv	1, 440	149.30	<0.0001
	Phen	4, 440	2.85	0.0235
	Dens	3, 440	715.17	<0.0001
	Cv × Phen	4, 440	1.00	0.4086
	Cv × Dens	3, 440	18.09	<0.0001
	Phen × Dens	12, 440	1.18	0.2945
	Cv × Phen × Dens	12, 440	0.71	0.7378

**Table 4 insects-15-00378-t004:** Results of GLMM (α = 0.05) on the percentage of berries with signs of *Botrytis cinerea* and sour rot in white and red cultivars.

Cultivar Category	Source of Variation	df	F Value	*p* Value
White cultivars	Cultivar (Cv)	1, 440	30.29	<0.0001
	Phenological stage (Phen)	4, 440	5.23	0.0004
	*H. halys* density (Dens)	3, 440	15.32	<0.0001
	Cv × Phen	4, 440	0.57	0.6865
	Cv × Dens	3, 440	4.30	0.0053
	Phen × Dens	12, 440	1.54	0.1055
	Cv × Phen × Dens	12, 440	0.94	0.5113
Red cultivars	Cv	1, 440	59.04	<0.0001
	Phen	4, 440	0.80	0.5232
	Dens	3, 440	1.17	0.3220
	Cv × Phen	4, 440	0.95	0.4367
	Cv × Dens	3, 440	0.80	0.4967
	Phen × Dens	12, 440	0.56	0.8752
	Cv × Phen × Dens	12, 440	0.70	0.7493

**Table 5 insects-15-00378-t005:** Results of GLMM (α = 0.05) on the mean weight of berries of white and red cultivars.

Cultivar Category	Source of Variation	df	F Value	*p* Value
White cultivars	Cultivar (Cv)	1, 440	343.27	<0.0001
	Phenological stage (Phen)	4, 440	1.41	0.2297
	*H. halys* density (Dens)	3, 440	1.17	0.3252
	Cv × Phen	4, 440	0.46	0.7132
	Cv × Dens	3, 440	0.46	0.7099
	Phen × Dens	12, 440	0.69	0.7628
	Cv × Phen × Dens	12, 440	0.98	0.4707
Red cultivars	Cv	1, 440	13.42	0.0003
	Phen	4, 440	1.79	0.1302
	Dens	3, 440	10.76	<0.0001
	Cv × Phen	4, 440	2.66	0.0322
	Cv × Dens	3, 440	1.22	0.3016
	Phen × Dens	12, 440	1.02	0.4259
	Cv × Phen × Dens	12, 440	1.01	0.4349

**Table 6 insects-15-00378-t006:** Results of GLMM (α = 0.05) on the glucose and fructose content of the must of white and red cultivars.

Sugar Content per Cultivar Category	Source of Variation	df	F Value	*p* Value
Glucose (white cultivars)	Cultivar (Cv)	1, 200	924.66	<0.0001
	Phenological stage (Phen)	4, 200	1.70	0.1515
	*H. halys* density (Dens)	3, 200	0.38	0.7661
	Cv × Phen	4, 200	6.88	< 0.0001
	Cv × Dens	3, 200	0.11	0.9558
	Phen × Dens	12, 200	0.93	0.5225
	Cv × Phen × Dens	12, 200	1.24	0.2554
Glucose (red cultivars)	Cv	1, 200	29.77	<0.0001
	Phen	4, 200	2.69	0.0321
	Dens	3, 200	0.88	0.4520
	Cv × Phen	4, 200	0.72	0.5797
	Cv × Dens	3, 200	1.75	0.1581
	Phen × Dens	12, 200	1.78	0.0541
	Cv × Phen × Dens	12, 200	1.39	0.1734
Fructose (white cultivars)	Cv	1, 200	530.37	<0.0001
	Phen	4, 200	1.17	0.3263
	Dens	3, 200	3.37	0.0195
	Cv × Phen	4, 200	4.82	0.0010
	Cv × Dens	3, 200	1.31	0.2714
	Phen × Dens	12, 200	1.24	0.2596
	Cv × Phen × Dens	12, 200	1.63	0.0860
Fructose (red cultivars)	Cv	1, 200	89.50	< 0.0001
	Phen	4, 200	3.30	0.0120
	Dens	3, 200	1.94	0.1237
	Cv × Phen	4, 200	0.72	0.5767
	Cv × Dens	3, 200	0.65	0.5832
	Phen × Dens	12, 200	2.16	0.0150
	Cv × Phen × Dens	12, 200	0.90	0.5462

**Table 7 insects-15-00378-t007:** Results of GLMM (α = 0.05) on the tartaric and gluconic acid content of the must of white and red cultivars.

Acid Content per Cultivar Category	Source of Variation	df	F Value	*p* Value
Tartaric acid (white cultivars)	Cultivar (Cv)	1, 195	182.05	<0.0001
	Phenological stage (Phen)	4, 195	16.03	<0.0001
	*H. halys* density (Dens)	3, 195	1.73	0.1612
	Cv × Phen	4, 195	14.02	<0.0001
	Cv × Dens	3, 195	3.23	0.0235
	Phen × Dens	12, 195	3.39	0.0002
	Cv × Phen × Dens	12, 195	1.95	0.0354
Tartaric acid (red cultivars)	Cv	1, 200	199.76	<0.0001
	Phen	4, 200	2.39	0.0523
	Dens	3, 200	5.95	0.0007
	Cv × Phen	4, 200	3.72	0.0061
	Cv × Dens	3, 200	0.75	0.5237
	Phen × Dens	12, 200	0.80	0.6509
	Cv × Phen × Dens	12, 200	1.01	0.4393
Gluconic acid (white cultivars)	Cv	1, 153	1.37	0.2439
	Phen	4, 153	5.71	0.0003
	Dens	3, 153	3.63	0.0145
	Cv × Phen	4, 153	1.51	0.2007
	Cv × Dens	3, 153	0.71	0.5482
	Phen × Dens	12, 153	2.19	0.0149
	Cv × Phen × Dens	12, 153	0.70	0.7449
Gluconic acid (red cultivars)	Cv	1, 157	0.44	0.5105
	Phen	4, 157	3.38	0.0110
	Dens	3, 157	4.86	0.0029
	Cv × Phen	4, 157	1.77	0.1373
	Cv × Dens	3, 157	1.82	0.1454
	Phen × Dens	12, 157	1.47	0.1416
	Cv × Phen × Dens	12, 157	0.74	0.7109

**Table 8 insects-15-00378-t008:** Results of GLMM model (α = 0.05) on *Halyomorpha halys* infestation observed in different treatments characterized by insecticide and fungicide applications using the beat sampling in Pinot gris vineyards in 2019.

Source of Variation	df	F Value	*p* Value
Treatment	5, 51.2	8.73	<0.0001
Sampling date (D)	3, 117	81.41	<0.0001
Treatment × D	15, 125	3.59	<0.0001

## Data Availability

The datasets generated during and/or analyzed during the current study are available from the corresponding authors on reasonable request.

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
