# Peer review of "Seasonal Dynamics and Damage of Halyomorpha halys in Italian Vineyards"

_insects, 2024, doi:10.3390/insects15060378_

Round 1

Reviewer 1 Report

Comments and Suggestions for Authors

The authors presented interesting results of seasonal dynamics of BMSB in vineyards. Overall, the MS is well written and organized, however I have some concerns regarding some topics:

In line 48-49: they mentioned that laboratory and semi-field studies indicated that nymphs are unable to complete their development when exclusively feeding on grapes. So, do adults only visit the grapes for feeding and not for reproduction? The reason why the infestation is only in the field borders (as you mentioned later in discussion).

-In materials and methods, why you collected once every two weeks in 2017 and weekly in weekly in 2018. In such kind of studies, the sampling must be done weekly.

Statistical analysis, I suggest to revise your analysis. Some parts are nor clear. More particularly, you have Zeros data so its important to fit an appropriate test. More details are mentioned in the attached pdf. In addition, three-way effects are quite difficult to interpret.

-Results: you have three factors, so focus only on the three-way interaction (if there is). Again, I am not a fan of three way factors.

you have a lot of tables. You can combine many tables in one table, for example tables (11, 12, 13, 14, 15 and 16) have the same factors, so you can put one table and you can identify the measured parameter in the left column. Idem for other tables.

You presented results as data not shown. These results are important and must be presented as figures.

What is the purpose of the figure in Supp Material?

Discussion is well written, but you can delete some lines that are nor necessary. However, you have to talk more about the impact of BMSB on Glucose and fructose and the relation with potential infection of fungi pathogen. This part is a major point.

Comments on the Quality of English Language

Reviewer 2 Report

Comments and Suggestions for Authors

This is an outstanding study, given the scope, careful experimentation, methods, and finally, the sheer volume of data provided to support the key conclusions. I count a total of 14 Figures and 19 Tables. The statistical analysis is also appropriate throughout (GLMM, Tukey's). In addition to documenting the phenology of H. halys on wine grapes in Italy, it is notable to see the results for several varieties. The key observation of an edge effect has been noted before in field crops (eg., soybean), but not as much for fruit crops. The results highlighted in the abstract are supported by the data. I therefore, see this paper an excellent data set that will be useful in designing additional studies with BMSB, as the pest continues to spread to new regions, countries. In addition, the English grammar is excellent, the refs cited are appropriate; I have no copy edits. The paper, if accepted, will receive broad interest. This is one of the few papers I have reviewed where it is Acceptable as-is; no revisions required.

Reviewer 3 Report

Comments and Suggestions for Authors

Comments For authors

M&M

Among the main observations, highlighting as deficiencies is the absence of a short description of at least the main pest management agricultural practices that are regularly applied in the vineyards studied (Conventional/IPM/Organic regime). It is  also missing to describe the main agricultural practices aimed at management of soil and herbaceous vegetation cover in the growing area.

L106. IInfestation densities instead of treatments.

Results

L193. “Beat sampling revealed a higher count of stink bugs if compared to visual sampling, with the latter consistently underestimating their numbers (Tables 1 and 2; Figures 1A and S1). You indicate that beat sampling allows for greater capture records of stink bugs compared to visual sampling, and you indicate that this can be verified in Tables 1 and 2, but I cannot see it from the values indicated in them. Could you be more explicit?  You also point out that it can be verified in Figure 1A, (and I suppose that if so, also in 1B), although in 1A the values represented refer only to the beat sampling. On the other hand, Supplementary Figure S1 lacks a graph footer so it does not allow us to know what type of sampling it refers to.

L195-196. The authors  report that in the beat samplings a variation in the values of H halys can be seen during the season (refer to Fig. 1A). I wonder: why don't they also include Fig 1B here? After all, it is also corresponding to beat sampling, although referring to the year 2018…? In any case, no Figures are provided where the data or the seasonal variation of visual sampling is presented...   

L196-198. “its infestation was also influenced by the grape cultivar (Table 2), resulting higher on Merlot and Pinot gris  compared to Cabernet Franc and Glera (Table 2; Figure 2A)”.  

In 2017 (Fig. 2A), statistically higher values are observed in the Merlot and Pinot gris cultivars compared to Cabernet Franc and Glera, although in Figure 2B (2018) the data are relatively different from those of 2017. It would be interesting to have of a possible explanation for this variation between both years (in the Discussion section).

L213-214. “During the 2018 growing season, H. halys population level fluctuated during the samplings  (Table 3; Figure 1B), increasing at the end of the season particularly in Cabernet Franc, Merlot and Raboso (Table 3; Figure 1B)” (a suggestion in view of Fig. 1B).

L214-216. “Overall, we observed differences among cultivars, with Cabernet Franc and Merlot showing the highest infestation levels, while Tai, Glera, Raboso and Cabernet Sauvignon the lowest (Figure 2B)”.  However, in 2017 the Cabernet Franc cultivar presented values lower than Merlot and Pinot Gris, and similar to Glera.

L 216. “Pinot gris experienced a moderate level of infestation”. The results between both years are relatively contradictory, since this statement corresponds to the year 2018 (Fig. 2B) although this is not the case in 2017 (Fig. 2A).

 L217-219. “Halyomorpha halys numbers were higher on red grape cultivars than on white ones (Figure 3A), and in late-ripening cultivars compared to early ones (Table 4; Figure 3B). Pest distribution along rows was influenced by the position, being higher at the border than at the center (Table 3; Figure 3C).” This result is compatible with a predominantly terrestrial dispersal of H. halys nymphs. The comparison between interior and edge zones is fully evident in Fig. 3C.

 L236-238. “Concerning white cultivars, the percentage of berries showing signs of infestation was different among cultivars (Table 5), higher on Glera than on Pinot Gris (Table 5; Figure 4), as well as while for red ones (Table 6), it was higher on Cabernet Franc than on Merlot (Table 6; Figure 5).

L238-241. The extent of damage to berries due to H. halys infestation varied across phenological stages in white cultivars (Table 5), while there was no significant difference observed for the red ones  (Table 6). Why do you state that in the red clusters there were no significant differences based on phenology? The p value was 0.0235. Please explain.

L241-244. “On white cultivars, a significant interaction between these two factors was also  detected (Table 5). In the case of these cultivars, Glera berries exhibited the highest damage when infestation occurred at 50% veraison, while no similar effect was observed for Pinot gris (Figure 4)”. OK

L244-246. In the case of red cultivars, the percentage of damaged berries per cluster considering grape cultivars showed a general increase when the infestation increased, with Cabernet Franc more susceptible to H. halys damage than Merlot (Figure 5). OK

L261-263. The percentage of berries with signs of B. cinerea due to brown marmorated stink bug infestations was different between white-berry cultivars (Table 7) but not for red-berry ones, and the incidence of this pathogen was higher on Pinot gris than on Glera (Table 7; Figure 6A).

L264-268. “On red cultivars, the percentage of berries with signs of B. cinerea and sour rot was different between cultivars (Table 8), with higher levels on Merlot than on Cabernet Franc (data not shown). On white cultivars, the presence of signs of grey mold and sour rot associated with H. halys infestation differed among phenological stages (Table 7), with a higher incidence following infestations from the grape ripening period (Figure 6B).

L279-283. An effect of brown marmorated stink bug infestation on the mean berry weight emerged on red cultivars but not on white ones (Tables 9 and 10). On red cultivars indeed, berry weight was higher in the case of 0.5 and 1 H. halys adults per cluster than in the other two treatments infestation densities (Figure 7A), with differences in the interaction cultivar*plant phenological stage (Figure 7B).

In Figure 7A there must be a mistake in the title of the ordinate axis, should it be “average berry weight”? while “Damaged berries (%)” should be in Figure 7B.

L 373-376. “Nevertheless, in the conditions under study we noticed a continuous occurrence of different developmental stages of the pest throughout the sampling period that suggests a certain degree of plant suitability.” In my opinion, this represents a great contribution on the knowledge  of ecology of H. halys.

Conclusions

In my opinion it is too similar to Discussion. In the Conclusions the main contributions of this study are somewhat mixed with results from studies reported by other authors, which does not allow this section to be clearly differentiated from the Discussion itself. It would be much more appropriate for the authors to present here only the conclusions, and in a more synthetic way, which would make it gain in performance. Information included here, such as the type of management that is applied in a standard way in the study area, is specific to Materials and Methods, and not in the Conclusions. Similarly, statements such as: “Moreover, the occurrence of B. cinerea and sour rot infections can be influenced by climatic conditions during the ripening period. At the time of the experiments, conditions were not particularly conductive to gray mold development… it should be moved to Discussion.

Round 2

Reviewer 1 Report

Comments and Suggestions for Authors

I read the MS for the 2d time. The authors have improved it, but I still have some comments:: “

1-In the rebuttal, it was mentioned the following:

It is known that nymphs are not able to complete their development when exclusively feeding on grapes, but we cannot say for sure that adults only visit the grapes for feeding and not for reproduction: this is an aspect that should be investigated by ad hoc studies. To retain this information (and the potential for feeding-related higher infestation at the field border), we added the following sentence to the Discussion: “The inability of the pest to complete its development solely by feeding on grapes [10,11] could have contributed to the increased infestation level at the border observed in the surveyed vineyards” (lines 395-398)”.

Since there is no research info regarding feeding behavior of larvae on grape, my suggestion is to modify the sentence in line 65, to be more clear that immature and adults feed only on host plants.

The sentence you added in L395-398 doesn’t give any sense, please rephrase it!

In your field survey, have you found any immatures stages? If no, be more concise.

Because of missing data of complete development on grapes, don’t overinterpret your results, about the impact of BMSB on grapes.

-The authors have used zero inflated log normal distribution. I don’t think it’s the right one, since more zero values than others. Zero inflated passion distribution might be better. Check it again.

-Discussion is still long. Consider to shorten it (you can delete the overinterpretation).

-I confused the authors in my previous comment in conclusion. I meant to delete the citations. Its your conclusion and no need for citation.

Comments on the Quality of English Language

th eMS Needs E editing
